# A Review on the Use of Anti-TNF in Children and Adolescents with Inflammatory Bowel Disease

**DOI:** 10.3390/ijms20102529

**Published:** 2019-05-23

**Authors:** Martine A. Aardoom, Gigi Veereman, Lissy de Ridder

**Affiliations:** 1Department of Paediatric Gastroenterology, The Erasmus MC Sophia Children’s Hospital, 3015 GD Rotterdam, The Netherlands; l.deridder@erasmusmc.nl; 2Department of Paediatric Gastroenterology and Nutrition, Kidz Health Castle UZ Brussels, Free University Brussels, B-1090 Brussels, Belgium; gveereman@gmail.com

**Keywords:** anti-TNF, biological, inflammatory bowel disease, Crohn’s disease, ulcerative colitis, paediatrics, children, adolescents

## Abstract

Inflammatory bowel disease (IBD) presents with disabling symptoms and may lead to insufficient growth and late pubertal development in cases of disease onset during childhood or adolescence. During the last decade, the role of anti-tumor necrosis factor (TNF) in the treatment of paediatric-onset IBD has gained more ground. The number of biologicals presently available for children and adolescents with IBD has increased, biosimilars have become available, and practices in adult gastroenterology with regards to anti-TNF have changed. The aim of this study is to review the current evidence on the indications, judicious use, effectiveness and safety of anti-TNF agents in paediatric IBD. A PubMed literature search was performed and included articles published after 2000 using the following terms: child or paediatric, Crohn, ulcerative colitis, inflammatory bowel disease, anti-TNF, TNF alpha inhibitor, infliximab, adalimumab, golimumab and biological. Anti-TNF agents, specifically infliximab and adalimumab, have proven to be effective in moderate and severe paediatric IBD. Therapeutic drug monitoring increases therapy effectiveness and safety. Clinical predictors for anti-TNF response are currently of limited value because of the variation in outcome definitions and follow-ups. Future research should comprise large cohorts and clinical trials comparing groups according to their risk profile in order to provide personalized therapeutic strategies.

## 1. Introduction

Crohn’s disease and ulcerative colitis (UC) present with chronic inflammation of the bowel, and are therefore referred to as inflammatory bowel disease (IBD). In 8–25% of cases, IBD is diagnosed during childhood or adolescence (paediatric IBD) [1,2]. The current hypothesis regarding the pathogenesis of paediatric IBD is that the combination of a genetic predisposition, microbial factors and a susceptibility of the immune system lead to an aberrant inflammatory immune response. Despite this hypothesis, there is still no understanding of the dramatic increase in incidence of paediatric IBD worldwide [3]. Similar to adults with IBD, at diagnosis, paediatric patients may present with abdominal pain, diarrhea, weight loss, fever or rectal bleeding. But in addition, the onset of disease in an early stage of life may lead to insufficient growth, late pubertal development and psychosocial problems [4].

The treatment for IBD first aims to induce remission of disease and secondly to maintain remission. Maintenance therapy consists of immunomodulators such as thiopurines or methotrexate. Treatment options to induce remission for paediatric IBD were limited to 5-aminosalicilates, exclusive enteral nutrition (EEN) and corticosteroids until recently, while in adult patients, agents inhibiting tumor necrosis factor alpha (TNF-α) were already established. Anti-TNF is one of the agents within the group of biologicals that was first approved for use in the treatment of IBD. Nowadays, a broader spectrum of biologicals is available. These agents are available to physicians after a strict manufacturing and market authorization process regulated by the Food and Drug Administration (FDA) and the European Medicine Agency (EMA). Biologicals that are currently reimbursed or to treat paediatric IBD or under study are listed in Table 1.

In contrast to adults, in children and adolescents, the anti-TNF agents infliximab (IFX) and adalimumab (ADA) are currently the only biologicals approved by the FDA or EMA for treatment of IBD. IFX is a monoclonal chimeric anti-TNF antibody (partly murine, partly human) that was first approved in adults in 1998. In 2006 it was authorized by the Food and Drug Administration (FDA) to treat Crohn’s disease (CD) in children and adolescents. This drug, commercialized as Remicade^®^, was the first anti-TNF agent that was approved for paediatric IBD and had significant impact on the practice in paediatric gastroenterology [5]. In 2012 adalimumab (ADA), a fully humanized monoclonal anti-TNF antibody, was officially approved for application in paediatric CD, but is still under study for paediatric UC. Certolizumab pegol, a monoclonal antibody to TNF-α which comprises the Fab portion of the antibody conjugated to a polyethylene glycol, is another anti-TNF agent that is being used off-label in paediatric patients in some countries. It has been shown to be effective in reducing symptoms of moderately to severely active Crohn’s disease in studies including adult patients who had insufficient response to conventional therapy [6,7]. Golimumab blocks soluble and transmembrane TNF-α and is comparable with IFX, except that it is fully human and given by subcutaneous (sc) injections instead of intravenous (iv) infusions. This agent is approved for treatment of moderate to severe UC in adults but in children with IBD it is only available off-label [8,9,10].

## 2. Aim

The aim of this study was to review the current evidence on the indications, judicious use, effectiveness and safety of anti-TNF agents in children and adolescents with IBD. The search was focused on the most recent literature, but included previously published guidelines and their associated papers as they are relevant for the current treatment strategies. As the indication and effectiveness of anti-TNF may depend on how anti-TNF is used and how treatment is monitored, studies assessing these topics were also included. 

## 3. Materials and Methods

A literature search was performed in PubMed. Articles published after 2000 and written in English were included. The keywords IBD, CD, UC, children, paediatric, anti-TNF, TNF alpha inhibitor, biological, infliximab, adalimumab and golimumab were used for this search. For this review, mainly studies including paediatric patients were selected. Because findings in adult studies are sometimes extrapolated for treatment of children and adolescents, relevant adult studies were included if paediatric studies were absent on this topic. This was especially the case in the section that describes predictors of the effectiveness of anti-TNF in order to point out the lack of and need for studies assessing predictive markers for the effectiveness of anti-TNF in paediatric IBD. References from the selected manuscripts were searched for additional relevant studies.

## 4. Results

### 4.1. When and How to Use Anti-TNF

#### 4.1.1. Indications and Effectiveness in Crohn’s Disease

In the last two decades, indications for the use of anti-TNF therapy in paediatric IBD have changed. According to the guidelines and reimbursement criteria, anti-TNF agents should be used for induction and maintenance in children with CD in case of chronically active disease despite immunomodulators, and steroid refractory disease, the so-called step-up strategy [5,11]. 

In the latest guideline by the European Society of Paediatric Gastroenterology, Hepatology and Nutrition (ESPGHAN), the use of an anti-TNF agent as primary treatment strategy, which is referred to as top-down approach, is recommended in paediatric CD patients with active perianal fistulising disease. In addition, based on consensus, the top-down strategy should be considered in children with CD suffering from extensive disease, significant growth retardation, deep ulcerations in the colon seen at endoscopy, severe osteoporosis and stenosing or penetrating disease at diagnosis [12]. A recent study in the Southampton Children’s Hospital showed a decreased surgical rate between 2007 and 2017 from 7.1% to 5.1%, respectively, which was most pronounced in patients with CD (8.9% vs. 2.3%). Although the resection rate in patients treated with anti-TNF therapy was not significantly different from those who were not, a multivariate regression analysis showed anti-TNF therapy prevalence per year was the only significant predictor associated with reduction in surgical resection rate [13].

There are no studies with paediatric CD patients that describe a head to head comparison of IFX to ADA. A retrospective study in 200 adults compared anti-TNF naïve patients treated with IFX and ADA after matching for indication, disease phenotype according to the Montreal classification, duration of disease and age at starting therapy. Steroid-free clinical response, defined as no hospitalization for exacerbation, no discontinuation of anti-TNF therapy, and no need for or dependency on steroids was assessed after 1 and 2 years of follow up. At both time points, response rates were not significantly different when comparing ADA with IFX (62% vs 65% after 1 year, 41% vs 49% after 2 years, respectively) [14]. These findings are in line with other studies and were recently confirmed in a propensity-score matched comparison in 632 adult CD patients, showing no significant difference in steroid-free remission rate when comparing IFX with ADA after one year in patients who had received other previous therapies (19.1% vs 27.7% respectively, *p* = 0.350) [15,16,17]. Furthermore, in addition to positive findings with IFX, ADA was recently proven to be effective in children and adolescents with moderately to severely active CD complicated by perianal fistulae in fistula closure [18,19]. In line with these findings, consensus-based guidelines suggest that in paediatric patients previously naïve to anti-TNF therapy both IFX or ADA can be offered, taking into account the availability, administration route, costs and patient preferences [12].

#### 4.1.2. Indications and Effectiveness in Ulcerative Colitis

Treatment guidelines state that in the treatment of paediatric UC patients, IFX should be considered in case of chronic disease activity or steroid-dependency that cannot be controlled by 5-ASA and thiopurines for both induction and maintenance therapy [20]. The effectiveness of IFX in inducing clinical remission and mucosal healing in UC patients has been shown in several adult and paediatric studies [11,21,22]. If IFX is not effective at the standard dose of 5 mg/kg in inducing remission the dose should be increased in order to optimize effectiveness [20]. A recent study in children with steroid refractory UC compared 73 children receiving an intensified induction dose (mean induction dose ≥7mg/kg or interval ≤5 weeks between doses 1 and 3) with 52 children who received standard dosing. Intensified induction was associated with a higher chance of remission (Hazard ratio (HR) 3.2, *p* = 0.02) and a lower chance of colectomy (HR 0.4, *p* = 0.05), which indicates that an intensified IFX induction might be beneficial in children with steroid refractory UC [23]. Current guidelines state that in case of loss of response or intolerance to IFX, ADA or golimumab should be considered [20].

In case of an acute severe colitis, a medical emergency in children, defined by a high clinical disease activity score (paediatric ulcerative colitis activity index; PUCAI) ≥65, IFX is recommended as second-line medical therapy for anti-TNF naïve children failing intravenous corticosteroids [24]. PUCAI scores at days 3 and 5 have been shown to yield the best validated predictive values, and should therefore form the basis for decision making on when to start IFX [25,26].

When it comes to paediatric IBD, the number of performed randomized controlled trials (RCT) is scarce. Most of the aforementioned recommendations in guidelines are based on observational studies or extrapolated from adult trials. RCTs involving placebo versus an anti-TNF agent for induction treatment in paediatric IBD patients are lacking and not the way to go anymore, since efficacy has been proven in adults extensively by now. It would not be ethical to randomise children to placebo, since no true equipoise exists against the active treatment [27]. RCTs for (extended) maintenance versus placebo could be considered in case an escape treatment arm is provided and patients are in clinical remission after induction therapy. Important RCTs in paediatric IBD during the last two decades, summarized in the current guidelines, concern the dosing and administration of anti-TNF and show the effectiveness of anti-TNF therapy in paediatric IBD (Table 2). 

#### 4.1.3. Indication and Effectiveness of Early Anti-TNF Use 

The RISK study recently showed in a propensity-score matched analysis (*n* = 68 per group) that early anti-TNF (within <3 months after diagnosis) was associated with higher corticosteroid- en surgery free remission rates at one year compared with early immunomodulator therapy [30]. Kugathasan et al. compared paediatric CD patients who received anti-TNF within 90 days of diagnosis and had a successful completion of induction doses and at least one maintenance dose, with those who received anti-TNF therapy at a later stage, in a prospective inception cohort study in the US and Canada. They found that patients with early anti-TNF therapy had a significantly lower risk of developing penetrating complications (HR 0.30, 95% CI 0.10–0.89, *p* = 0.03). For the development of stricturing complications, no significant difference was found [31]. A comparison of top-down with step-up treatment in a South Korean cohort found that deep remission and mucosal healing rates were higher in the top-down group [32,33]. Although these findings are promising, studies are limited by the non-randomized study design. Data from future risk stratification studies and RCTs are needed for more specific and evidence-based statements on indications for top-down therapy. Anti-TNF has been shown to be effective in the majority of patients. Ideally, we should be able to prescribe it only for patients who will respond favourably.

#### 4.1.4. The Indication and Effectiveness of Biosimilars

Since the expiration of the patent for infliximab in 2015, a number of biosimilars have obtained EMA and FDA approval and are used in clinical practice. The FDA defines a biosimilar as a biological product that is highly similar to the reference product with respect to safety, purity and potency. The aim in biosimilar development is to demonstrate similarity to the originator in specific conditions and therefore needs to be studied in both in vitro and ex vivo assays. Initially, efficacy, safety and immunogenicity were reported similar for biosimilar CT-P13 (Remsima^®^) and the originator (Remicade^®^) in 2 multicenter double-blind randomized phase I and phase III studies in patients with ankylosing-spondylitis and rheumatoid arthritis [34,35]. Based on the concept of extrapolation, approval of CT-P13 following the phase III trial included indications of IBD and paediatric IBD. Most knowledge on the safety and efficacy of biosimilars in IBD patients is based on studies in adult patients. A recent systematic review included data from 11 observational studies. Meta-analysis comprising 552 mostly adult IBD patients treated with CT-P13 showed high rates of clinical response and disease remission that sustained over 1-year. The risk of adverse events was similar in patients treated with CT-P13 compared to the risk reported in patients treated with the originator [36]. A few studies in paediatric IBD patients described findings following induction therapy with a biosimilar of IFX, and found similar efficacy for clinical response or remission. These studies described small cohorts and follow-up data were limited to 14 weeks [37,38,39]. In a prospective study, 39 paediatric IBD patients who were in remission or had mild disease activity switched from the IFX originator to Remsima^®^ during maintenance therapy. No serious adverse events occurred and none of the patients had a disease exacerbation during the mean follow-up period of 8 months [40]. One other study compared 38 patients who switched to CT-P13 with 36 patients maintained on the IFX originator. After one year of follow up 77.8% and 78.9% of paediatric IBD patients had been in persistent remission, respectively. No statistically significant differences were found for pharmacokinetics, immunogenicity and number of adverse events [41]. The position paper by the Paediatric IBD Porto Group of ESPGHAN states that switching from the originator to a biosimilar may be considered in case of clinical remission and after induction, but multiple switches (>1) are not recommended because data on interchangeability is limited and it compromises traceability of the drug [42]. In 2017 biosimilars for ADA became available. All currently available biosimilars are listed in Table 3.

So far, there are no available data on the safety and efficacy of ADA biosimilars in paediatric IBD patients. For all available and future biosimilars it is strongly recommended to collect sufficient post-marketing surveillance data on efficacy, safety and immunogenicity [42].

#### 4.1.5. Prediction of Anti-TNF Responsiveness 

Anti-TNF treatment failure may occur due to primary non-response, diminished response or loss of response (secondary non-response) or adverse drug reactions. Certain clinical characteristics are known to predict response to anti-TNF therapy, mainly based on studies with adult patients. For both UC and CD factors associated with a good response are younger age (<40 years) at diagnosis, concomitant use of an immunomodulator and being naïve to anti-TNF therapy [15,43,44,45]. In addition, shorter disease duration [45], isolated colonic disease [46], elevated CRP [47,48], the absence of previous surgery in CD [15] and a hemoglobin >11.5 mg/dL in UC [49] are considered to have predictive value [50]. Conversely the following predictors of primary non-response to anti-TNF therapy have been reported: IBD patients with severe disease and high BMI [51,52], UC patients with low serum albumin and low haemoglobin at anti-TNF initiation [53,54] and CD patients with fibrostenotic disease [55], previous intestinal resection and a disease duration of more than 2 years [44,56]. In a prospective cohort including 995 CD patients (PANTS study) obesity, smoking, low albumin concentrations, higher baseline markers of disease activity and development of immunogenicity were all associated with low drug concentrations during induction resulting in non-remission at week 54 following anti-TNF therapy. This suggests that part of the non-response to anti-TNF might be resolved by increasing the target drug concentration during induction [57]. Although this is one of the few studies assessing response to anti-TNF in which children and adolescents (≥6 years) are included, no subanalysis has been performed for this group so far. 

Recently the therapeutic aim for CD has shifted from symptom control to mucosal healing, hence preventing the development of stricturing or penetrating disease. There may be a window of opportunity allowing early treatment to prevent further bowel damage since there is evidence that alterations of the immune response occur years before diagnosis [58]. Therefore, detecting preclinical disease with specific biomarkers may help prediction of therapy responsiveness. Some candidate biomarkers may be found in the genetic field. Arijs et al. compared pre-treatment colonic mucosal expression profiles of refractory UC patients who responded to IFX therapy (defined as complete endoscopic and histological healing) to the non-responders to IFX therapy. Seventy-four probe sets were found, representing 53 known genes. The top 5 of differentially expressed genes were osteoprotegerin (TNFRSF11B), stanniocalcin-1 (STC1), prostaglandin-endoperoxide synthase 2 (PTGS2), interleukin 13 receptor alpha 2 (IL13Ralpha2) and interleukin 11 (IL11). Together these genes predicted the response to IFX with 89% accuracy. All of the proteins encoded by these genes are involved in the adaptive immune response [59]. In a recent study, Bank et al. aimed to replicate previous findings [60,61] in a new cohort of 587 CD and 458 UC patients and to find new single nucleotide polymorphisms (SNPs) associated with anti-TNF response. Although the results should be confirmed in other cohorts, they indicate that polymorphisms in genes involved in the regulation of the NFĸB pathway, the TNF-α signaling pathway and other cytokine pathways are associated with response to anti-TNF therapy [62]. West et al. describes an overexpression of the cytokine oncostatin M (OSM), which correlates closely with histopathological disease severity, in inflamed intestinal tissue from mice and humans, particularly in patients with anti-TNF resistant disease [63]. Currently, studies regarding genetic profiling, metabolomics and microbiome are ongoing to enable the prediction of IBD disease course and response to therapy. Future studies are needed to validate previous findings. 

## 5. Combination Therapy

Although 60–87% of therapy refractory patients initially respond to anti-TNF induction therapy, 23–46% of primary responders lose anti-TNF response over time, showing a 31–40% loss of response rate in paediatric patients receiving monotherapy [64,65,66]. The most important contributor to loss of response is immunogenicity. Because anti-TNF agents consist of large and complex proteins, the formation of anti-TNF antibodies is triggered. The combination of anti-TNF with an immunomodulator such as azathioprine, 6-mercaptopurine, or methotrexate may prevent loss-of-response due to reduced immunogenicity. The SONIC trial was the first RCT in biologic and immunomodulator naïve patients that showed that adult CD patients receiving combination therapy had superior clinical and endoscopic outcomes compared to the patients on IFX monotherapy [50]. Patients receiving combination therapy had higher drug levels and lower IFX antibody levels. In 2014 an RCT including adult patients with moderate to severe UC treated with IFX demonstrated similar findings. IFX combined with azathioprine was superior to monotherapy with azathioprine or IFX, while there was no superiority of IFX monotherapy over azathioprine [67]. Also, for adalimumab, combination therapy was superior over monotherapy reflected by better response rates, drug survival and a decreased number of hospitalizations and abdominal surgeries [17].

In children and adolescents, the European guideline on paediatric CD states that there is insufficient evidence to define the risk-benefit ratio for mono- or combination therapy [12], but since the publication of this guideline several studies in children have shown for both CD and UC that combination therapy lowers the risk of antibody formation [68,69,70]. Kansen et al. studied 229 children with CD and found a lower probability of remaining free of antibodies to infliximab (ATI) in the group of children who received IFX monotherapy compared to children receiving combination therapy at 12, 24 and 36 months (72.6% vs 93.4%, 57.7% vs 91% and 48.1% vs 91%, respectively). Moreover, the incidence of ATI formation was significantly lower in children receiving continuous combination therapy (*p* = 0.003) as was in children receiving early combined combination therapy, until a median duration of 6.2 months (*p* = 0.008) compared to monotherapy [68]. This is in line with findings from an RCT including 99 paediatric CD patients that compared the efficacy and safety of maintenance therapy with ongoing combination therapy to IFX monotherapy after 26 weeks of combination therapy. No significant differences were documented between groups for clinical response, disease activity scores and endoscopic findings at 54 weeks. The need for treatment intensification or modification was comparable in both groups [71]. In a prospective observational study in 37 paediatric CD patients, mucosal healing was evaluated in patients receiving monotherapy or combination therapy with IFX or ADA. No significant differences were found for complete mucosal healing but combination therapy was superior for complete and partial mucosal healing taken together (*p* < 0.01) [72]. 

The more recent European guidelines for treatment of paediatric UC recommend induction therapy with IFX in combination with an immunomodulator. After 6 months, discontinuation of the immunomodulator may be considered, especially in boys [20]. Temporary combination therapy is recommended due to the risk for lymphomas, in particular the lethal hepatosplenic T-cell lymphoma (HSTCL), that occurs more often in young male patients. Concerns regarding the development of malignancies when using combination therapy are justifiable according to a prospective registry (DEVELOP registry) including 5766 paediatric IBD patients with 25,543 patient-years of follow-up (PYF) and showing a significantly higher standardized incidence ratio (SIR) of 3.06 (95% CI 1.32–6.04) for malignancies in patients who received combination therapy with a biological and thiopurine compared to 1.11 (95% CI 0.03–6.16) in patients with biologic monotherapy. When no stratification for thiopurines was performed, no significantly higher incidence rates were found in patients receiving a biological, providing a good reason to recommend discontinuation of the immunomodulator [73]. 

Prior to discontinuation of the immunomodulator one should optimize the dosage in order to obtain IFX trough levels >5mg/mL. A retrospective study including 223 adult CD patients showed that patients with adequate through levels fared well after immunomodulator withdrawal. Thirty-eight percent of patients needed IFX dose increase after withdrawal of the immunomodulator and 18% discontinued IFX [74]. The same authors showed that for similar effectiveness, combination therapy reduces IFX drug consumption and that IFX doses need to be increased upon discontinuation of the immunomodulator [75].

## 6. Safety of Anti-TNF Agents

A large population-based study in paediatric IBD patients (*n* = 9442) compared to the general population showed a 3-fold increased mortality risk with a HR of 6.6 (95% CI 5.3–8.2) when it comes cancer as a cause of mortality. Most frequently reported cancer types in paediatric IBD are colorectal carcinomas, cholangiocarcinomas and lymphomas, specifically the hepatosplenic T cell lymphomas [76,77]. The latter is a feared complication due to its fatality, but so far, no relationship with the use of biologics or anti-TNF in specific has been reported. A systematic review including 36 adult patients with HSTCL reported no cases on anti-TNF monotherapy [78,79]. A Swedish cohort study including 9405 paediatric IBD patients reviewed the occurrence of cancer between groups with different drug exposures, including anti-TNF, but found no significant differences between groups [77].

The second most frequent cause of death in all paediatric IBD patients, as described by Olen et al., was digestive diseases (*n* = 54, HR 36.8, 95% CI 21.3–67.6, including IBD) followed by infections (*n* = 6, HR 6, 95% CI 2.1–16.9). They found that the relative risk for death has not decreased with development of new drugs for treatment of IBD, such as anti-TNF [80]. This study was underpowered to directly assess the effect of biologics on mortality but a systematic review on the risk of serious infection in paediatric IBD patients on anti-TNF therapy showed a similar risk in patients on anti-TNF therapy compared to the expected rate of serious infection with immunomodulator therapy in paediatric patients (333 per 10,000 PYF; SIR, 1.06; *p* = 0.65; 95% CI 0.83–1.36). The rate of serious infections in the included prospective studies was similar between ADA and IFX (294 per 10,000 PYF vs 357 per 10,000 PYF, respectively; incidence rate ratio 0.82; *p* = 0.46; 95% CI 0.46–1.37) [81]. They did find a significantly lower risk for serious infections compared to paediatric IBD patients treated with steroids. The previously described large DEVELOP cohort by Hyams et al. compared incidence rates of malignancy and hemophagocytic lymphohisticytosis (HLH), a disorder of immune hyperstimulation and dysregulation that is associated with fatal consequences, in paediatric IBD patients exposed to IFX with patients not exposed to biologics. IFX exposure was not associated with increased risk of malignancy (SIR 1.69; 95% CI 0.46–4.32). The 5 cases of HLH registered in this cohort all occurred in patients using thiopurines, none of those patients had exposure to IFX or ADA [73]. 

Besides this study, no other large studies were performed that assessed the role of anti-TNF on the risk of these rare but severe complications in paediatric IBD. A large study in 190,694 adults with IBD did confirm that combination therapy was associated with increased risks of serious infection (HR 1.23; 95% CI 1.05–1.45) compared to anti-TNF monotherapy. In addition, compared to thiopurine monotherapy, anti-TNF monotherapy was associated with an increased risk of serious infection (HR 1.71; 95% CI 1.56–1.88) but on the other hand with a decreased risk of opportunistic viral infection (HR 0.57; 95% CI 0.38–0.87), which shows the heterogeneity of findings [82]. Although it should be stressed that absolute risks are small in all the aforementioned studies and the number of available studies is limited, the findings suggest that patients with paediatric IBD should be followed closely with regard to disease activity, treatment and risk of these complications. 

## 7. Therapeutic Drug Monitoring and When to Exit

Considering the variability in the pharmacokinetics of anti-TNF agents among IBD patients, therapeutic drug monitoring (TDM) is required to obtain optimal serum concentrations for effectiveness. Several studies, in both adult UC and CD patients, have shown that the use of TDM during anti-TNF therapy improves clinical outcomes and reduces antibody formation [83,84,85,86,87]. For paediatric UC, the European guideline therefore recommends measuring drug levels and anti-drug antibody levels following induction in order to optimize treatment. In addition, measuring drug levels is useful in the assessment of unsatisfactory response to anti-TNF to guide dose escalation or a switch to another biologic [20]. It is known that TNF levels are influenced by multiple factors, including disease severity and the degree of intestinal inflammation [88,89]. This justifies intensified dosing in children with acute severe colitis. Drug levels obtained during induction maximize efficacy [24,90].

The optimal timing for the use of TDM in anti-TNF treatment for IBD patients is still debatable. Whether TDM should be performed in a proactive manner, by measuring serum drug levels at pre-specified time points, or in a reactive manner in case of loss of response, remains unclear. The first RCT to compare adjusted drug dosing based on trough levels (proactive) with dosing based on clinical activity (reactive) was the TAXIT trial. At enrolment trough levels were highly variable in these adult IBD patients and optimized to reach a target trough level prior to optimization. Concentration-based dosing was not superior to clinically-based dosing in achieving remission after 1 year [91]. The subsequent Tailorix study investigated dose adjustment based on symptoms, biomarker analysis and/or serum concentration compared to dose adjustment based on symptoms alone in adult CD patients receiving IFX combination therapy in a prospective randomized exploratory trial. There were no significant differences in corticosteroid-free remission rates after 54 weeks [92]. In contrast, a retrospective study of 102 IBD patients reported that proactive TDM was independently associated with less treatment failure in a multivariate analysis (HR 0.15; 95% CI 0.05–0.51; *p* = 0.002) and fewer IBD-related hospitalizations (HR 0.18; 95% CI 0.05–0.99; *p* = 0.007) [93]. So far, no paediatric data have been reported comparing proactive and reactive TDM strategies. Singh et al. did show in 58 paediatric IBD patients that week 14 IFX levels were predictive for persistent remission at week 54 [69]. In addition, van der Hoeve et al. studied 35 children with IBD and found IFX trough levels just before the first maintenance infusion to be significantly higher in children achieving clinical and/or biological remission at week 52 [94]. These data suggest that reaching optimal trough levels during induction and prior to maintenance therapy improves the efficacy of anti-TNF.

Withdrawal of anti-TNF therapy could be considered in cases of sustained remission, although it may seem counterintuitive due to the fear of relapse or loss of efficacy. Reasons to consider withdrawal of anti-TNF are related to safety, side effects, costs or patient preferences. The STORI trial was the first study to assess the risk of relapse after discontinuation of anti-TNF therapy in adults. Patients had to be in steroid-free remission for at least 6 months while on at least 1 year of scheduled IFX combined with immunomodulators. A relapse rate of 43.9% after one year and 52.2% after two years was reported in the 115 CD patients [95]. Several prospective and retrospective studies have followed assessing this topic. A systematic review and meta-analysis including 27 studies showed an overall risk of relapse after discontinuation of anti-TNF of 44% in CD (95% CI 36–51%; I^2^ = 79%; 912 patients) and 38% for UC (95% CI 23–52%; I^2^ = 82%; 266 patients) [96]. Amongst others, factors predictive of relapse in IBD are elevated inflammatory markers (e.g. elevated leukocyte count, elevated C-reactive protein, elevated faecal calprotectin) and absence of mucosal healing [97]. Although no official guidelines are available on the discontinuation of anti-TNF agents in paediatric IBD, according to the available literature it is suggested to evaluate all clinical parameters and perform an endoscopy to assess mucosal healing prior to withdrawal of anti-TNF therapy. Furthermore, in children and adolescents with IBD growth and pubertal development should be a priority when considering discontinuation of therapy. 

## 8. Gaps in Knowledge and Future Perspectives

Current management of paediatric IBD with regard to anti-TNF therapy is illustrated in Figure 1, which shows an example of anti-TNF therapy strategy for a specific patient. 

The future place of anti-TNF therapy in the treatment of paediatric IBD will depend strongly on the role of newly developed agents. Other biological agents are already used to treat refractory paediatric IBD, vedolizumab being the most widely used. Currently available data find this monoclonal antibody acting against α4β7-integrin to be safe and effective, while it has a slow induction rate and seems less effective in CD patients compared to UC patients [98,99,100,101]. A systematic review and meta-analysis recently showed that vedolizumab, together with IFX, was ranked highest for induction of clinical remission when compared to anti-TNF agents, and janus kinase (JAK) inhibitors in UC adult patients. In addition, vedolizumab was considered safest in terms of serious adverse events and infection [102]. Because only small cohorts of paediatric IBD patients using vedolizumab are described and the role of vedolizumab in therapy-naïve paediatric IBD patients is unclear, larger prospective trials to assess efficacy and safety of vedolizumab are needed. Other biological agents that are currently being investigated for their use in paediatric UC or CD are etrolizumab and ustekinumab (Table 1); the latter has been found to be effective in inducing a clinical response in CD patients who have failed or are intolerant to conventional treatments or TNF agents in phase III trials, and has also shown to be beneficial in numerous real-world observational studies in adults with refractory CD [103,104,105]. In children and adolescents with CD only one retrospective study in 44 children is available, showing a clinical remission rate of 38.6% after 12 months. Future prospective studies should confirm whether this agent is a viable alternative in the treatment of paediatric IBD [106].

Considering the limited amount of data regarding these relatively new therapies, IFX, together with ADA in case of CD, currently remains the most important biological agent for the treatment of paediatric IBD. However, future changes in treatment strategies can be expected including subcutaneous administration of IFX and due to the expected availability of other therapeutic agents.

## 9. Conclusions

Anti-TNF agents, specifically IFX and ADA, have proven to be effective in children and adolescents with moderate and severe IBD. Fortunately, it has been possible to limit the use of corticosteroids in this vulnerable population. However, it is imperative to carefully assess clinical indicators and disease behaviour for the prescription of anti-TNF therapy. In addition, costs and patient preferences play a role when weighing treatment options. 

Considering the large heterogeneity between paediatric IBD patients, it should be stressed that every patient should be evaluated separately. Overtreatment should be avoided, and for optimal therapy effect, TDM should be performed. During therapy, the patient should be closely monitored to prevent infections and other complications. For non-responders to anti-TNF therapy despite adequate trough level, alternative treatment modalities should be sought, which is challenging, since reimbursed options are currently limited. Clinical predictors for anti-TNF response are currently of limited value because of the variation in outcome definitions and follow-up. Importantly, data regarding specific biomarkers for paediatric IBD that could be used in daily clinical practice are lacking. More large cohorts and clinical trials comparing groups according to their risk profile are needed in order to provide safer and personalized therapeutic strategies for young patients.

## Figures and Tables

**Figure 1 ijms-20-02529-f001:**
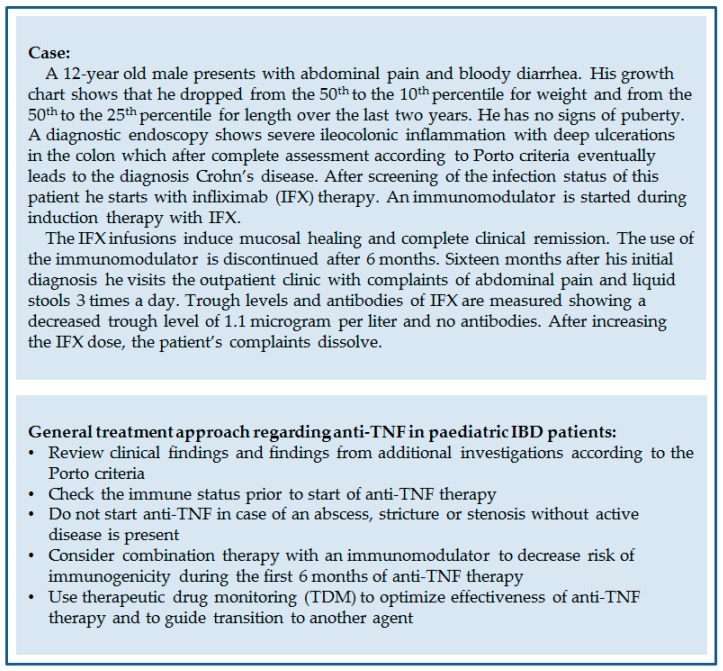
Treatment with anti-TNF in a specific patient.

**Table 1 ijms-20-02529-t001:** Biologicals that are currently reimbursed or under study for treatment of paediatric IBD.

Class	Name	Product (^®^)	Admission Route
Anti-TNF	infliximab	Remicade	iv
	adalimumab	Humira	sc
	golimumab	Simponi	sc
	certolizumab pegol	Cimzia	sc
Anti-α4β7integrin	vedolizumab	Entyvio	iv
Anti-α4β7 and αEβ7 integrin	etrolizumab	-	sc
Interleukin 12/23 p40 inhibitor	ustekinumab	Stelara	iv/sc

Abbreviations: CD, Crohn’s disease; UC, ulcerative colitis; iv, intravenous; sc, subcutaneous.

**Table 2 ijms-20-02529-t002:** Randomized controlled trials in paediatric IBD assessing how to use anti-TNF.

Study and Study Group	Agent	Indication	*N*	Study Aim	Definition of Outcome	Time Point	Response	Remission
Hyams 2007 [5] *REACH*	IFX	CD patients with a PCDAI >30	103	Comparison of IFX maintenance intervals: every 8 vs. every 12 weeks. Randomization took place after 10 weeks of IFX treatment.	**Response:** 15 point decrease in PCDAI **Remission:** PCDAI ≤10	Week 10	88%	59%
						Week 54	8 weeks group: 56%* 12 weeks group: 24%* (*p* = 0.001) **of week 10 responders*	8 weeks group: 56% 12 weeks group: 24% (*p* = 0.001)
Ruemmele 2009 [28] *GFHGNP*	IFX	CD	40	Comparison of scheduled IFX maintenance dosing every 8 weeks vs. IFX on demand. Randomization at week 10.	**Remission:** Harvey Bradshaw Index <5	Week 10	-	85%
						Week 60	-	Scheduled IFX: 83% IFX on demand: 61% (*p* = 0.001)
Hyams 2012 [29] *IMAgINE*	ADA	Moderate to severe CD	188	High dose ADA (40 mg or 20 mg for body weight ≥40 kg or <40 kg; *n* = 93) or low dose (20 mg or 10 mg for body weight ≥40 kg or <40 kg; *n* = 95). Randomization after 4 weeks.	**Response:** Decrease in PCDAI ≥15 **Remission:** PCDAI ≤10	Week 26	High dose: 59%* Low dose: 48%* (*p* = 0.073) **of patients with clinical response at week 4*	High dose: 39%* Low dose: 28%* (*p* = 0.075) **of patients with clinical response at week 4*
						Week 54	High dose: 42%* Low dose: 28%* (*p* = 0.038) **of patients with clinical response at week 4*	High dose: 33%* Low dose: 23%* (*p* = 0.100) **of patients with clinical response at week 4*
Hyams 2012 [11] *T72 study group*	IFX	UC	60	Comparison of IFX maintenance intervals: every 8 vs. every 12 weeks. Randomization took place after 8 weeks of IFX treatment.	**Response:** decrease in Mayo score by ≥30% and ≥3 points **Clinical remission:** Mayo score ≤2 with no individual subscore >1 and PUCAI <10	Week 8	73%	33%
						Week 54	-	8 weeks group: 38%* 12 weeks group: 18%* (*p* = 0.146) **of week 8 responders*

Abbreviations: CD, Crohn’s disease; UC, ulcerative colitis; IFX, infliximab; ADA, adalimumab; PCDAI, paediatric Crohn’s disease activity index; PUCAI, paediatric ulcerative colitis activity index.

**Table 3 ijms-20-02529-t003:** Biosimilars that are currently registered for the treatment of paediatric IBD.

INFLIXIMAB (Originator; Remicade)			ADALIMUMAB (Originator; Humira)		
Name	Manufacturer	Year of Registration	Name	Manufacturer	Year of Registration
Inflectra (CTP13)	Hospira	2013	Amgevita	Amgen	2017
Remsima	Celltrion/Egis	2013	Cyltezo	Boehringer Ingelheim	2017
Flixabi	Samsung Bioepis	2016	Imraldi	Samsung Bioepis	2017
Zessly	Novartis/Sandoz	2018	Hyrimoz	Novartis/Sandoz	2018

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
