# Peer review of "A Review on the Use of Anti-TNF in Children and Adolescents with Inflammatory Bowel Disease"

_ijms, 2019, doi:10.3390/ijms20102529_

Round 1

Reviewer 1 Report

The manuscript entitled “How and when to use anti-TNF in children and adolescents with inflammatory bowel disease” presents interesting issue, but it requires some important corrections.

Major:

1.      The manuscript is shabbily prepared (e.g. leadings, References section, tables, etc.)

2.      Moreover, the manuscript is unorganized. Authors should formulate their aim and afterwards present the issue comprehensively, but they should briefly present the issue only, not all information that they know.

3.      The manuscript presents rather kind of technical report, not a scientific publication, as it is not prepared with any structure. In spite of a fact that Authors prepared not research but a review article, their manuscript must have its structure (introduction – aim – materials and methods – results (literature review) – conclusions), so Authors must precisely specify what was the aim of their review, what results did they observe in their literature search.

General:

Instead of PIBD, Authors should rather use a term “pediatric IBD”, as rather IBD abbreviation is commonly used.

Title:

Title should be formulated rather as a title for a scientific review, not as a title for a column of a newspaper (asking a question as a title is not recommended)

Abstract:

In spite of a fact that Authors prepared not research but a review article, their Abstract must have its structure (justification – aim – results – conclusions), so Authors must precisely specify what was the aim of their review, what results did they observe in their literature search and what was their conclusion based on indicated results. In the present manuscript there is justification only.

Background:

This section is extremely shabbily prepared.

In a paragraph each sentence should “follow” the previous one, while in this section they seem as just random information.

The Introduction section should precisely justify the need for such analysis.

Lines 28-30 – this sentence is not associated with the following sentences

Information about pathogenesis should be presented before information about therapy

Aim of the study should be briefly presented

Materials and Methods:

Taking into account, that the Materials and methods section is not presented (it should be added), without any specific information, it is hard to understand which studies were included into review and why. Authors did not present any key words, which were used during literature search, inclusion and exclusion criteria of references, information about the procedure of literature search conducted by them, number of chosen references, as well as information if some of them were excluded from the review and on the basis of which criteria. As a number of publications that are not related to the issue were included (including the own publications of Authors), it is a serious problem.

The serious flaw of the presented manuscript is associated with the fact, that it presents a highly subjective review, not a systematic review. While the systematic review has a key role for broadening knowledge, the other reviews don’t have such role.

Main body of the manuscript:

It should be extensively corrected, as Authors have very specific aim – to indicate how and when to use anti-TNF in pediatric patients.

Authors should not refer only situation in Netherlands and Belgium (e.g. table 1), as they do not want to publish their manuscript in a local journal, but in an international one. They should at least present EU data and USA data, if they are not able to gather more information.

Conclusions:

More specific information should be presented to answer the questions about how and when. Bullet points specifying detailed strategy for a specific patients would be recommended.

Author Response

Reviewer #1

The manuscript entitled “How and when to use anti-TNF in children and adolescents with inflammatory bowel disease” presents interesting issue, but it requires some important corrections.
We thank the reviewer for acknowledging this topic being of interest.

Point 1. Major:
Point 1a. The manuscript is shabbily prepared (e.g. leadings, References section, tables, etc.)
Response 1a. We are sorry to read this comment and suspect that the reviewer expects a systematic review?

Point 1b. Moreover, the manuscript is unorganized. Authors should formulate their aim and afterwards present the issue comprehensively, but they should briefly present the issue only, not all information that they know.

Response 1b. We thank the reviewer for pointing this out. We acknowledge that the title of the manuscript might have been misleading and does not fully cover the aim of this review. We aim to provide a comprehensive review on the use of anti-TNF in children and adolescents with IBD. This includes how and when to use this agent, but cannot be reviewed without also describing recent updates on the use of combination therapy, therapeutic drug monitoring and the role of biosimilars. Our intention is not to address a research question for a specific patient phenotype. The topic is much broader. We therefore structured the manuscript this way. In order to clarify we adjusted the title and clarified our aim, as will also be described in Response 3 and 5.

Point 1c. The manuscript presents rather kind of technical report, not a scientific publication, as it is not prepared with any structure. In spite of a fact that Authors prepared not research but a review article, their manuscript must have its structure (introduction – aim – materials and methods – results (literature review) – conclusions), so Authors must precisely specify what was the aim of their review, what results did they observe in their literature search.

Response 1c. We thank the reviewer for pointing this out and acknowledge that in case of a systematic review, we should indeed have set up the manuscript as suggested. We aimed to write a comprehensive review on the use of the use of anti-TNF in pediatric IBD based on the most important findings in scientific literature. We prepared this manuscript based on the guidelines presented on the website of the journal IJMS, which state that a review manuscript should include the front matter (title, author list, affiliations, abstract and keywords) and the back matter (supplementary materials, acknowledgments, author contributions, conflicts of interest and references). By our knowledge we have adhered to the guidelines of IJMS. We have also performed a thorough literature search on the subject. It was not our aim to write a systematic review and we therefore were not able to change the set-up of the manuscript as the reviewer suggests. Following this comment we have better clarified our aim and the justification of our aim in the background section.

Point 2. General: Instead of PIBD, Authors should rather use a term “pediatric IBD”, as rather IBD abbreviation is commonly used.

Response 2. Following this comment we replaced ‘’PIBD’’ by ‘’paediatric IBD’’ in the entire manuscript.

Point 3. Title: Title should be formulated rather as a title for a scientific review, not as a title for a column of a newspaper (asking a question as a title is not recommended)

Response 3. We thank the reviewer for pointing this out. We changed the title to ‘’A comprehensive review on the use of anti-TNF in children and adolescents with inflammatory bowel disease’’.

Point 4. Abstract: In spite of a fact that Authors prepared not research but a review article, their Abstract must have its structure (justification – aim – results – conclusions), so Authors must precisely specify what was the aim of their review, what results did they observe in their literature search and what was their conclusion based on indicated results. In the present manuscript there is justification only.

Response 4. Following this comment we rewrote the last sentences in our abstract to clarify our aim. We aimed to write a comprehensive review on the use of the use of anti-TNF in pediatric IBD based on the most important findings in scientific literature. As we did not perform a systematic review with specific research questions, we were not able to structure our abstract as such. In our response to Point 6 we explain this further.

Point 5. Background: This section is extremely shabbily prepared.

Point 5a. In a paragraph each sentence should “follow” the previous one, while in this section they seem as just random information.

Response 5a. We thank the reviewer for pointing this out. We rewrote the background section in order to create a fluent introduction (page 2, lines 33 – 58).

Point 5b. The Introduction section should precisely justify the need for such analysis.

Response 5b. Following this comment we added sentences to the background section (page 2, lines 47 – 58).

Point 5c. Lines 28-30 – this sentence is not associated with the following sentences

Response 5c. We agree with the reviewer that this should be improved. We rewrote the background section as a whole and changed and replaced this sentence.

Point 5d. Information about pathogenesis should be presented before information about therapy

Response 5d. We agree with the reviewer and therefore moved this sentence within the background section.

Point 5e. Aim of the study should be briefly presented

Response 5e. We thank the reviewer for pointing this out. We added a few sentences in the background section following this comment (page 2, lines 52 – 58).

Point 6. Materials and Methods:
Taking into account, that the Materials and methods section is not presented (it should be added), without any specific information, it is hard to understand which studies were included into review and why. Authors did not present any key words, which were used during literature search, inclusion and exclusion criteria of references, information about the procedure of literature search conducted by them, number of chosen references, as well as information if some of them were excluded from the review and on the basis of which criteria. As a number of publications that are not related to the issue were included (including the own publications of Authors), it is a serious problem. The serious flaw of the presented manuscript is associated with the fact, that it presents a highly subjective review, not a systematic review. While the systematic review has a key role for broadening knowledge, the other reviews don’t have such role.

Response 6. We would like to refer to our response to Point 1c. In addition, in this review we incorporated all major publications related to this topic by our knowledge. We based our manuscript on the present guidelines and newer publications regarding the specific topic if available. It was not our intention to include publications not related to the issue and we are not aware of any such references. If the reviewer could specify these references, we would be happy to review these and consider removal. Also, in case the reviewer knows major publications related to paediatric IBD currently missing in this manuscript, we would be happy to add those. Since the authors have contributed to guidelines and work in the field, it is inevitable that their work is referred to. The authors were asked to provide a comprehensive review because they are experts in this field.

Point 7. Main body of the manuscript:
Point 7a. It should be extensively corrected, as Authors have very specific aim – to indicate how and when to use anti-TNF in pediatric patients.

Response 7a. In the main body of the text we indeed try to provide an update on how and when to use anti-TNF in paediatric IBD patients. Therefore, in a large part of our manuscript, we focus on this topic (lines 183 – 337) specifically. But in order to know how and when to use anti-TNF, issues regarding combination therapy, therapeutic drug monitoring, the role of biosimilars and the future perspective are pivotal and are therefore also described in this manuscript. We acknowledge that the title of our manuscript might have been misleading and did not cover the complete aim of the manuscript. We therefore changed the title of the manuscript.

Point 7b. Authors should not refer only situation in Netherlands and Belgium (e.g. table 1), as they do not want to publish their manuscript in a local journal, but in an international one. They should at least present EU data and USA data, if they are not able to gather more information.

Response 7b. We thank the reviewer for pointing this out. We acknowledge that the columns describing status of the agents are only applicable on specific national situations. We have removed the columns describing the reimbursement status of these agents (Table 1, page 3) and adjusted the corresponding sentences (page 2, line 64-66).

Point 8. Conclusions:
More specific information should be presented to answer the questions about how and when. Bullet points specifying detailed strategy for a specific patients would be recommended.

Response 8. As this comment is related to previous comments about the title and aim, we refer to our response to Point 1 and 5. As there is a lot of variation in phenotype of paediatric IBD patients, providing a detailed strategy for a specific patient in the conclusion is not feasible.

Reviewer 2 Report

This is a valuable and practical paper and I would strongly support its publication.

There are three points which should be addressed:

The statement: "The pathogenesis of PIBD is currently explained by a combination of a 41 genetic predisposition, microbial factors and a susceptibility of the immune system leading to an aberrant inflammatory immune response." is a rather optimistic interpretation of our knowledge. Although we have some understanding of the pathophysiology of IBD we still have no understanding of its cause and dramatic increase in incidence worldwide. 

Table 1 needs to be modified. Either the section on reimbursement should be removed or extended with examples of countries where there is and is not reimbursement. In its present form it is largely reflecting the situation in Belgium and the Netherlands

The statements "a significantly higher standardized incidence ratio (SIR) of 3.06 (95% CI 1.32-6.04) for  malignancies in patients who received combination therapy with a biological and thiopurine  compared to 1.11 (95% CI 0.03-6.16) in patients with biologic monotherapy. However, in a subanalysis the SIR was not significantly higher for IFX and thiopurines compared to IFX 274 monotherapy" need clarification as, on the face of it, they are contradictory.

Author Response

Reviewer #2

This is a valuable and practical paper and I would strongly support its publication.

We thank the reviewer for appreciating our work.

There are three points which should be addressed:
Point 1. The statement: "The pathogenesis of PIBD is currently explained by a combination of a 41 genetic predisposition, microbial factors and a susceptibility of the immune system leading to an aberrant inflammatory immune response." is a rather optimistic interpretation of our knowledge. Although we have some understanding of the pathophysiology of IBD we still have no understanding of its cause and dramatic increase in incidence worldwide.

Response 1. We thank the reviewer for bringing this to our attention. Following this comment we changed the sentence to ‘’The current hypothesis regarding the pathogenesis of PIBD is that the combination of a genetic predisposition, microbial factors and a susceptibility of the immune system lead to an aberrant inflammatory immune response’’ (page 2, line 36). In addition, we added a sentence and reference regarding our lack of understanding of the dramatic increase in incidence of pediatric IBD worldwide (page 2, line 39).

Point 2. Table 1 needs to be modified. Either the section on reimbursement should be removed or extended with examples of countries where there is and is not reimbursement. In its present form it is largely reflecting the situation in Belgium and the Netherlands

Response 2. We thank the reviewer for pointing this out. We acknowledge that the columns describing status of the agents are only applicable to specific national situations. We therefore have removed the columns describing the reimbursement status of these agents (Table 1, page 3) and adjusted the corresponding sentences (page 2, line 65).

Point 3. The statements "a significantly higher standardized incidence ratio (SIR) of 3.06 (95% CI 1.32-6.04) for malignancies in patients who received combination therapy with a biological and thiopurine compared to 1.11 (95% CI 0.03-6.16) in patients with biologic monotherapy. However, in a subanalysis the SIR was not significantly higher for IFX and thiopurines compared to IFX 274 monotherapy" need clarification as, on the face of it, they are contradictory.

Response 3. We fully agree with the reviewer that these statements seem contradictory. The statements are based on Figure 2 in the study by Hyams et al. (Gastroenterology 2017) in which the findings are stratified by thiopurine use and shown for biologicals and IFX separately. In a supplementary table the authors show the SIRs without stratification by thiopurine use and find no significant difference for both the infliximab group and the biologics group. For our review the main message is that when not stratifying for thiopurine use, no significantly increased SIR for malignancies is seen when using anti-TNF or biologicals. Therefore, we removed the contradicting sentence from the manuscript. In order to clarify our statement in the manuscript we also added the following sentence: ‘’When no stratification for thiopurines was performed no significantly higher incidence rates were found in patients receiving a biological, pointing out the reason to recommend discontinuation of the immunomodulator’’ (page 3, line 388).

Round 2

Reviewer 1 Report

The manuscript entitled “A comprehensive review on the use of anti-TNF in children and adolescents with inflammatory bowel disease” presents interesting issue, but it still requires a number of important corrections, as some issues were not corrected.

Major:

1.      Authors corrected mainly the title and almost nothing more – they indicated in their title that they present “comprehensive review”, but in fact their manuscript is not a comprehensive review. Comprehensive review can not present a random unorganized data that are somehow associated with the issue. It must have its rules and structure. If Authors want to prepare the comprehensive review, they should get familiar with some comprehensive reviews, e.g. https://www.ncbi.nlm.nih.gov/pubmed/23306415, https://www.ncbi.nlm.nih.gov/pubmed/20594090 and afterwards prepare their comprehensive review according the standards.

2. The manuscript is still shabbily prepared (e.g. leadings, References section, tables, etc.) – Authors should address the instructions for authors and prepare the manuscript accordingly.

3. Moreover, the manuscript is unorganized. Authors should formulate their aim and afterwards present the issue comprehensively, but they should briefly present the issue only, not all information that they know. It was not corrected.

4. The manuscript presents rather kind of technical report, not a scientific publication, as it is not prepared with any structure. In spite of a fact that Authors prepared not research but a review article, their manuscript must have its structure (introduction – aim – materials and methods – results (literature review) – conclusions), so Authors must precisely specify what was the aim of their review, what results did they observe in their literature search.

Title:

Title should be corrected to correspond the real content of the manuscript – Authors indicated in their title that they present “comprehensive review”, but in fact their manuscript is not a comprehensive review.

Abstract:

In spite of a fact that Authors prepared not research but a review article, their Abstract must have its structure (justification – aim – results – conclusions), so Authors must precisely specify what was the aim of their review, what results did they observe in their literature search and what was their conclusion based on indicated results. In the present manuscript there is justification only.

Background:

Authors present all the information in only one paragraph, so section is hard to follow.

Materials and Methods:

Taking into account, that the Materials and methods section is not presented (it should be added), without any specific information, it is hard to understand which studies were included into review and why. Authors did not present any key words, which were used during literature search, inclusion and exclusion criteria of references, information about the procedure of literature search conducted by them, number of chosen references, as well as information if some of them were excluded from the review and on the basis of which criteria. As a number of publications that are not related to the issue were included (including the own publications of Authors), it is a serious problem.

The serious flaw of the presented manuscript is associated with the fact, that it presents a highly subjective review, not a systematic review. While the systematic review has a key role for broadening knowledge, the other reviews don’t have such role.

Main body of the manuscript:

It should be extensively corrected, as Authors have very specific aim – to indicate how and when to use anti-TNF in pediatric patients.

Conclusions:

More specific information should be presented to answer the questions about how and when. Bullet points specifying detailed strategy for a specific patients would be recommended.

Author Response

Reviewer #1

The manuscript entitled “A comprehensive review on the use of anti-TNF in children and adolescents with inflammatory bowel disease” presents interesting issue, but it still requires a number of important corrections, some issues were not corrected.

Point 1. Authors corrected mainly the title and almost nothing more – they indicated in their title that they present “comprehensive review”, but in fact their manuscript is not a comprehensive review. Comprehensive review cannot present a random unorganized data that are somehow associated with the issue. It must have its rules and structure. If Authors want to prepare the comprehensive review, they should get familiar with some comprehensive reviews, e.g. https://www.ncbi.nlm.nih.gov/pubmed/23306415, https://www.ncbi.nlm.nih.gov/pubmed/20594090

and afterwards prepare their comprehensive review according the standards.

Response 1. We thank the reviewer for pointing this out. According to this comment we removed and adjusted major parts of the manuscript.
1) Removed previous sections ‘’2. Available agents’’ (page 3) and ‘’3.2. Increasing use of anti-TNF in paediatric IBD’’ (page 8). in order to make this review more comprehensive and to the point.
2) Moved and shortened previous section ‘’5. Biosimilars’’ to organize the manuscript in
accordance with the aim of this review (page 13).
3) Removed a large part of previous section ‘’5. Safety of anti-TNF agents’’ in order to focus on the safety issues related to the use of anti-TNF specifically (page 11).
4) We removed the word comprehensive from the title as according to the examples provided by the reviewer a comprehensive review is more of a systematic review. We acknowlegde that this is not a systematic review and therefore removed the word ‘’comprehensive’’ from the title. We do feel that with the substantial changes to the manuscript, which we will futher elaborate on in this letter, we wrote a structured and thorough review.

Point 2. The manuscript is still shabbily prepared (e.g. leadings, References section, tables, etc.) – Authors should address the instructions for authors and prepare the manuscript accordingly.
Response 2. We adjusted the titles of the tables in the manuscript in order to fulfill the requirement to have a short and explanatory title and changed all tables to the same color. Furthermore, by our knowledge we followed the exact format and instructions that were found on https://www.mdpi.com/journal/ijms/instructions. We have carefully reviewed the manuscript once again, but feel that this is line with the instructions. We uploaded a pdf document of the new ‘’clean’’ version to illustrate what the word document should look like. We also uploaded a pdf version and word version of the ‘’track changes’’ document. In case any other adjustments need to be made in order to address the instructions for authors we would be happy to change this after receiving specific guidance on this.

Point 3. Moreover, the manuscript is unorganized. Authors should formulate their aim and afterwardspresent the issue comprehensively, but they should briefly present the issue only, not all informationthat they know. It was not corrected.

Response 3. We have made several adjustments in order to present the issue more comprehensively.

1) We stuctured the manuscript by adding the subheading ‘’Aim’’ and formulated the aim of this review more clearly (page 3, lines 116 – 122).
2) We organized the manuscript in the way the reviewer suggests in the next point (please see point 4).
3) We removed substantial parts of the manuscript in order to present the issue only (as
described in our response to point 1).
4) Several sentences throughout the whole manuscript were added to formulate the findings in the results section in relation to our aim and present the issue comprehensively.

Point 4. The manuscript presents rather kind of technical report, not a scientific publication, as it is not prepared with any structure. In spite of a fact that Authors prepared not research but a review article, their manuscript must have its structure (introduction – aim – materials and methods – results (literature review) – conclusions), so Authors must precisely specify what was the aim of their review, what results did they observe in their literature search.
Response 4. Following this comment and the suggestion of the editor, we now structured the manuscript as suggested (introduction – aim – materials and methods – results (literature review) – conclusions).

Point 5. Title should be corrected to correspond the real content of the manuscript – Authors indicated in their title that they present “comprehensive review”, but in fact their manuscript is not a comprehensive review.
Response 5. We changed the title following this comment. The more detailed response regarding this comment was discussed in Response 1.4.

Point 6. Abstract: In spite of a fact that Authors prepared not research but a review article, their Abstract must have its structure (justification – aim – results – conclusions), so Authors must precisely specify what was the aim of their review, what results did they observe in their literature search and what was their conclusion based on indicated results. In the present manuscript there is justification only.
Response 6. This structure is now incorporated in the abstract. As we wanted to adhere to the guidelines and author instructions of the journal, we did not include the specific headings itself in the abstract, but the abstract now includes the structure the reviewer suggests.

Point 7. Background: Authors present all the information in only one paragraph, so section is hard to follow.
Response 7. Following this comment we now completely rewrote the background section – which is now named ‘’Introduction’’ in accordance with the structure the reviewer suggested (page 2, line 62 till page 3, line 114). The introduction now first describes the disease, followed by information on the available treatment strategies for this disease and anti-TNF specifically.

Point 8. Materials and Methods: Taking into account, that the Materials and methods section is not presented (it should be added), without any specific information, it is hard to understand which studies were included into review and why. Authors did not present any key words, which were used during literature search, inclusion and exclusion criteria of references, information about the procedure of literature search conducted by them, number of chosen references, as well as information if some of them were excluded from the review and on the basis of which criteria. As a number of publications that are not related to the issue were included (including the own publications of Authors), it is a serious problem.
Response 8. We incorporated a Materials and Methods section as suggested by the reviewer (page 3). This section includes a search strategy, keywords that were used and inclusion and exclusion criteria. As this was not a systematic review, we were not able to describe the number of chosen references. With the removal of several sentences in the part about ‘’Safety of anti-TNF agents’’ (page 11, line 541), we also removed several references that the reviewer may have considered not to be related to the issue. If there are more, we would like to ask the reviewer to specify the included publications that are not related to the issue, so we can review these and consider removal.

Point 9. The serious flaw of the presented manuscript is associated with the fact, that it presents a highly subjective review, not a systematic review. While the systematic review has a key role for broadening knowledge, the other reviews don’t have such role.
Response 9. We appreciate the value of a systematic review. Next to systematic reviews, we do think thorough reviews also have added value, especially to update readers on quite specific topics, such as the use of anti-TNF in children and adolescents with Inflammatory Bowel Diseases.

Point 10. Main body of the manuscript: It should be extensively corrected, as Authors have very specific aim – to indicate how and when to use anti-TNF in pediatric patients.
Response 10. We rewrote a considerable part of the main body of the manuscript following this point. We elaborated on this in Response 1. and Response 3.

Point 11. Conclusions: More specific information should be presented to answer the questions about how and when. Bullet points specifying detailed strategy for a specific patients would be recommended.
Response 11. Following this comment we included a case of a specific patient treated with anti-TNF, providing a detailed strategy for this specific patient. In addition we also provided important general points of attention when treating paediatric IBD patients with anti-TNF (Figure 1, page 13 line 683 -804).

Round 3

Reviewer 1 Report

The manuscript entitled “A review on the use of anti-TNF in children and adolescents with inflammatory bowel disease” presents interesting issue, but it still requires some corrections, as some issues were not corrected.

General:

The manuscript is still shabbily prepared (e.g. numbering sections, tables, etc.) – Authors should address the instructions for authors and prepare the manuscript accordingly.

Authors should avoid personal forms (e.g. “our aim”) and they should use rather not personal ones (e.g. “the aim”).

Author Response

Reviewer #1 The manuscript entitled “A review on the use of anti-TNF in children and adolescents with inflammatory bowel disease” presents interesting issue, but it still requires some corrections, as some issues were not corrected.

General: Point 1. The manuscript is still shabbily prepared (e.g. numbering sections, tables, etc.) – Authors should address the instructions for authors and prepare the manuscript accordingly.

Response 1. Following this comment we contacted the assistant editor, who pointed out that we needed to renumber all sections, starting with ‘’1. Introduction’’. In addition we had to adjust the journal abbreviations in the references section by adding a dot after the journal name abbreviations. Both adjustments were made and can be found in the attached manuscript.

Point 2. Authors should avoid personal forms (e.g. “our aim”) and they should use rather not personal ones (e.g. “the aim”).

Response 2. We thank the reviewer for pointing this out. Based on this suggestion all sentences including personal forms were rewritten. We adjusted sentences on page 1 (line 18-20), page 3 (line 80-85 and 88-93) and page 8 (line 463).